# Teaching Research Ethics to Pharmacists: The Practice of Participatory Learning

**DOI:** 10.3390/pharmacy8040179

**Published:** 2020-09-28

**Authors:** Miku Ogura, Rieko Takehira, Etsuko Arita

**Affiliations:** Department of Medical Psychology, Pharmaceutical Education Research Center, Kitasato University School of Pharmacy, Tokyo 108-8641, Japan; dp18203@st.kitasato-u.ac.jp

**Keywords:** pharmacy education, workshop, clinical research, ethical education

## Abstract

The research history of community pharmacists in Japan is short, and ethical responses may not be mature. Therefore, the Japan Pharmaceutical Association and universities are working on research ethics education to help pharmacists make appropriate ethical responses. In this study, we evaluated whether an educational program using participatory learning was effective in research ethics education for pharmacists. Regarding the educational effects of our workshop, the score for motivation to learn about research ethics was high, and that for judgment and applied skills related to research ethics was low. Overall, participants’ assessment of the program contents was extremely favorable, indicating their satisfaction. Participatory learning was widely accepted and suggested to be effective in improving learning motivation. Additionally, to be able to apply the knowledge of research ethics to own research, it was considered necessary to continue learning through participatory learning. This will help pharmacists gain judgment and applied skills related to research ethics.

## 1. Introduction

Recently, pharmacists have been increasingly engaged in clinical research. Ethical considerations are necessary because the subjects of these studies include patients. Researchers are required to obey not only the four basic principles of medical ethics (respect for autonomy, non-maleficence, beneficence, and justice) [1] but also the laws and guidelines set by each country. The Council for International Organizations of Medical Sciences (CIOMS) has recommended that researchers and members of research ethics committees should undergo a basic training program on human research ethics [2]. In Japan, researchers are required to obtain education and training in research ethics [3].

However, the research history of community pharmacists in Japan is short, and ethical responses may not be mature. Therefore, the Japan Pharmaceutical Association and universities are working on research ethics education to help pharmacists make appropriate ethical responses [4]. Although lectures and e-learning methods [4,5] are widely used in Japan as research ethics educational programs, lately, the focus has been on participatory learning as a new educational method. In this method, learners participate in educational programs and promote mutual learning through group activities [6,7]. This method is widely applied overseas in research ethics education [8,9,10].

Therefore, we designed a workshop (WS) using participatory learning as an educational program. At the WS, discussions were held based on cases of ethical issues in clinical research. As part of this research, we conducted a preliminary study of community pharmacists’ willingness to learn. The results showed that they wanted the learning experience to be fulfilling [11]. Therefore, the WS included familiar cases that stimulated the intellectual curiosity of pharmacists, leading to a fulfilling learning experience. This study aimed to evaluate the educational effectiveness of the WS and to help establish a research ethics participatory educational program for future pharmacists.

## 2. Materials and Methods

We conducted a pre- and post-questionnaire survey with 45 pharmacists who participated in our WS on 13 October 2019. The pre-questionnaire survey investigated participants’ backgrounds, their reason for participating in the WS, etc. The post-questionnaire survey used a five-point Likert scale (1 = Strongly disagree, 5 = Strongly agree) and covered topics such as the educational effects of the WS and general assessment of the WS. Questionnaire data were analyzed using IBM SPSS Statistics 26 descriptive statistics.

This study was approved by the Kitasato Institute Hospital Research Ethics Committee (Approval number: 19045).

## 3. Results

### 3.1. Demographic Characteristics of the Participants

A total of 45 pharmacists participated in the WS. The response rate for the pre-questionnaire survey was 97.8% (*n* = 44), and that for the post-questionnaire survey was 93.3% (*n* = 42). Table 1 presents the demographic information of the participants, comprising 52.3% women, in a wide age range, possessing a BS degree (79.5%), and working in the area of community pharmacy (84.1%).

### 3.2. Participants’ Educational Experience

Table 2 shows participants’ educational experience in research ethics. A total of 59.1% of the participants had received research ethics education in the past, mostly after employment (61.5%). Additionally, numerous participants selected “lectures” (69.2%) and “e-learning” (42.3%) as the method of learning.

### 3.3. Reasons for Participating in the WS

We asked the participants their reason for attending the WS. A total of 28 pharmacists wanted to be able to judge whether the research required ethical approval from the Research Ethics Committee. Further, 23 pharmacists wanted to be able to judge research that fell under “Medical and Health Research Involving Human Subjects” and 23 pharmacists wanted to learn about research ethics through case studies.

### 3.4. Educational Effects of the WS on Participants

Table 3 presents the average scores for the various effects of the WS on the participants. The following items scored high: motivation for research ethics education (Item 7) scored 4.2 (SD = 0.8); the indispensability of research ethics education (Item 8) scored 4.6 (SD = 0.5), and motivation for future WS participation (Item 9) scored 4.5 (SD = 0.6). The following items scored low: learning connected to one’s research (Item 2) scored 3.9 (SD = 0.6); judgment as to whether the research falls under “Medical and Health Research Involving Human Subjects” (Item 4) scored 3.8 (SD = 0.8); judgment of application to the Research Ethics Committee (Item 5) scored 3.9 (SD = 0.8); and utility for future research (Item 6) scored 3.8 (SD = 0.7).

### 3.5. Participants’ Assessment of the WS

Figure 1 shows the participants’ assessment of the WS content and their overall satisfaction. While all the items received a high assessment score, free remarks in the discussion (Item 4) and overall satisfaction (Item 6) were strongly agreed to, or agreed to, by all the participants.

## 4. Discussion

This study investigated whether educational programs using participatory learning were effective in teaching research ethics to pharmacists.

Nearly half (40.9%) of the participants were beginners, with no prior research ethics education. Most of those who had received research ethics education had received it after employment, either through their organization’s educational program or through self-study. However, educational content at pharmacies and hospitals is not defined in the university curriculum [12]; therefore, the educational content may be different. The survey results suggested that the participants had different educational experiences and backgrounds and had access to different educational content. Moreover, most of those who had received research ethics education received it through lectures and e-learning. These methods are highly convenient and helpful for acquiring knowledge [13,14,15,16,17,18,19]. Thus, participants were suggested to be in an environment where they could easily acquire knowledge of research ethics. However, it is difficult to learn research ethics exclusively from knowledge. Regarding the reasons for WS participation, many participants answered that they wanted to make ethical judgments and conduct ethical case studies, suggesting that they participated to experience practical learning. Laws and guidelines on research ethics contain only basic rules and cannot resolve all ethical dilemmas [20]. Our previous studies also indicated that community pharmacists may become confused when encountering research situations that they cannot judge by themselves [21]. This suggests that the participants felt their judgment on research ethics was inadequate and wanted to learn about ethical issues through concrete cases.

Regarding the educational effect of the WS, the willingness to learn about research ethics for the future (Items 7 to 9) was high, and it was clear that the participants’ motivation to learn was high after participating in the WS. Case studies have reported that learners can imagine a situation in which a problem occurs, which leads to an increase in learning motivation [22,23]. Therefore, it is important to incorporate case studies in workshops as it augments the motivation to learn about research ethics. However, the scores for the acquisition of judgment on research ethics (Items 4 and 5) and connection with their own research (Items 2 and 6) were low. Participants felt it was difficult to apply knowledge on research ethics to their own research. In Japan, community pharmacists are rarely in clinical research as part of their everyday practice. Therefore, familiarity with clinical research varies depending on the individual pharmacist’s experience and environment. This may have affected their judgment and applied skills. The effectiveness of the following two approaches for practical learning research ethics has been reported. First, participatory learning with cases of ethical dilemmas has a positive impact on learners’ appropriate ethical decision-making [24]. Second, by participating in an educational program consisting of multiple lectures and learning about research ethics extensively, learners can become confident in researching on their own [25,26]. This WS was implemented as a participatory learning method using cases with strong educational effects. However, it was a single event. Therefore, it is presumed that the educational effects were insufficient, and it was difficult for the participants to gain adequate judgment and skills after just one WS. Therefore, it is suggested that pharmacists need to continue learning through participatory learning to be able to apply the knowledge of research ethics to their own research.

Overall, the participants’ assessment of the program contents was highly positive; they were extremely satisfied. Participants had diverse educational backgrounds, suggesting that this WS educational program could be widely applied to pharmacists. In addition, among the program contents, the score for discussion (Item 4) was high. The exchange of opinions during discussions deepens the knowledge and understanding of ethical issues [27,28]. Therefore, it is considered that this WS, which included discussions, deepened the participants’ understanding, and it was received positively by many participants.

## 5. Conclusions

We incorporated case studies and discussions for participatory learning at the WS. This learning method was widely accepted, and it was suggested that it could be effective from the viewpoint of improving the motivation to learn. Additionally, it is suggested that pharmacists need to continue learning through participatory learning to be able to apply the knowledge of research ethics to their own research. It is expected that the results of this study will contribute to a better understanding of research ethics education for pharmacists.

## Figures and Tables

**Figure 1 pharmacy-08-00179-f001:**
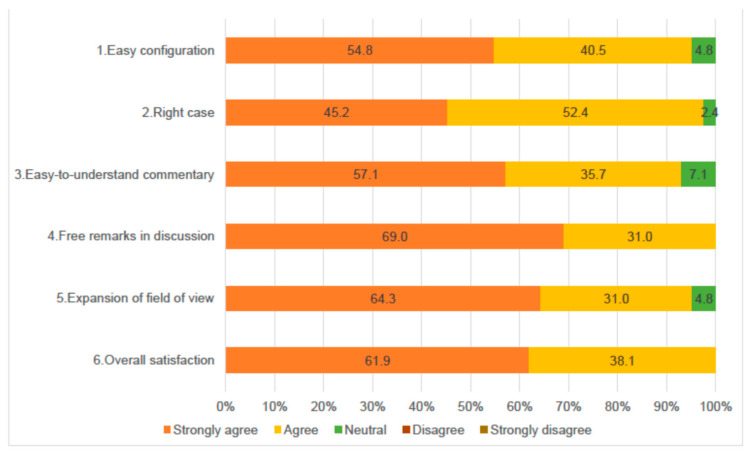
Participants’ WS assessment (*n* = 42).

**Table 1 pharmacy-08-00179-t001:** Demographic characteristics of the participants (*n* = 44).

Variable		No. (%)	
Gender			
	Male	18	(40.9)
	Female	23	(52.3)
	No response	3	(6.8)
Age			
	20–29	2	(4.5)
	30–39	12	(27.3)
	40–49	10	(22.7)
	50–59	14	(31.8)
	≥60	6	(13.6)
Academic degree			
	BS	35	(79.5)
	MS, PhD	8	(18.2)
	No response	1	(2.3)
Current employer			
	Community pharmacy	37	(84.1)
	Hospital	4	(9.1)
	Pharmacy school	1	(2.3)
	Others	1	(2.3)
	No response	1	(2.3)

**Table 2 pharmacy-08-00179-t002:** Participants’ experience of research ethics education.

Question			No. (%)	
1. Have you ever received research ethics education?				
(*n* = 44)				
	Yes		26	(59.1)
	No		18	(40.9)
If you answered “Yes” to Question 1, (*n* = 26)				
	1.1. When did you learn about research ethics? ^a^			
		University days	3	(11.5)
		Graduate school days	2	(7.7)
		After employment at current organization	16	(61.5)
		Others	6	(23.1)
	1.2. How did you learn about research ethics? ^a^			
		Lecture	18	(69.2)
		E-learning	11	(42.3)
		Participatory learning	3	(11.5)
		Others	1	(3.8)

^a^ Multiple responses were allowed.

**Table 3 pharmacy-08-00179-t003:** Educational effects of the WS on the participants (*n* = 42).

**No.**	**Item**	**Mean (SD)**
1	I was able to clarify my doubts	4.1 (0.5)
2	I was able to learn by connecting with my research	3.9 (0.6)
3	I was able to discover the ethical issues in research	4.0 (0.7)
4	I was able to judge research that fell under “Medical and Health Research Involving Human Subjects”	3.8 (0.8)
5	I was able to judge whether a study needed ethical approval from the Research Ethics Committee	3.9 (0.8)
6	I thought I could employ this knowledge in future research	3.8 (0.7)
7	I wanted to learn more about “research ethics”	4.2 (0.8)
8	I thought it was essential for pharmacists to know about research ethics	4.6 (0.5)
9	I wanted to participate in such a workshop again	4.5 (0.6)

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
