# Peer review of "Teaching Research Ethics to Pharmacists: The Practice of Participatory Learning"

_pharmacy, 2020, doi:10.3390/pharmacy8040179_

Round 1

Reviewer 1 Report

This paper is clearly written and contains foundational information that is important. It clearly identifies a problem that requires intervention and recommends a necessary component of that interventions. I encourage the authors to research various participatory educational interventions and publish those results to increase the ethics training so needed. 

Author Response

Point: This paper is clearly written and contains foundational information that is important. It clearly identifies a problem that requires intervention and recommends a necessary component of that interventions. I encourage the authors to research various participatory educational interventions and publish those results to increase the ethics training so needed.

Response: Thank you very much for providing important comments. We are thankful for the time and energy you expended. I brushed up my English. This paper has been proofread by a native speaker. In the future, we would like to contribute to research ethics education in pharmacy by studying various participatory educational interventions and publish the results. Thank you for your valuable advice.

Reviewer 2 Report

This is nice piece of report about methodological approach about research ethics, used in workshop for community pharmacists.

Research work is not everyday practice in community pharmacy and participants are most probably not representative samples of them. But all are aware of ethical questions. It would be nice to read something about relation among professional ethic of pharmacist and research ethics. These are indeed overlapping but anyway two different point of view. Individuals differently comprehend ethics of profession and of research, especially if they are not involved in research. The results presented in 3.4. Educational effects may be the consequence of individual comprehension the ethics or at least influenced by it. Participants who are not (or are less) research oriented may higher score case studies (for everyday practice) and lower judgment of research applications. The comment of authors supported by answer from the survey would be strongly appreciated.

Similarly, there is missing more intensive comment about "judgement of ..." (Items 4 and 5). Namely, judgement of applications and research fulfilment of criteria have two different faces: one is very technical even managerial, another is deeply ethical. Beginners, as such participants were declared, often hardly distinguish among them. Comment would be useful.

The results of 3.5. assessment… are quite expected, since participants were motivated—they had chosen to participate at the workshop. This is just observation, not criticism of the manuscript, but the fact could be mentioned.

Author Response

Point: Research work is not everyday practice in community pharmacy and participants are most probably not representative samples of them. But all are aware of ethical questions. It would be nice to read something about relation among professional ethic of pharmacist and research ethics. These are indeed overlapping but anyway two different point of view. Individuals differently comprehend ethics of profession and of research, especially if they are not involved in research.

The results presented in 3.4. Educational effects may be the consequence of individual comprehension the ethics or at least influenced by it. Participants who are not (or are less) research oriented may higher score case studies (for everyday practice) and lower judgment of research applications. The comment of authors supported by answer from the survey would be strongly appreciated.

Similarly, there is missing more intensive comment about "judgement of ..." (Items 4 and 5). Namely, judgement of applications and research fulfilment of criteria have two different faces: one is very technical even managerial, another is deeply ethical. Beginners, as such participants were declared, often hardly distinguish among them. Comment would be useful.

The results of 3.5. assessment… are quite expected, since participants were motivated—they had chosen to participate at the workshop. This is just observation, not criticism of the manuscript, but the fact could be mentioned.

Response: Thank you very much for providing important comments. We are thankful for the time and energy you expended. Thank you for understanding this research and giving us useful advice. We agree with you and have incorporated this suggestion our paper. We added the sentence "In Japan, community pharmacists are rarely in clinical research as part of their everyday practice. Therefore, familiarity with clinical research varies depending on the individual pharmacist's experience and environment. This may have affected their judgment and applied skills." to line 125 of the Discussion.